# Years of Schooling Could Reduce Epigenetic Aging: A Study of a Mexican Cohort

**DOI:** 10.3390/genes12091408

**Published:** 2021-09-13

**Authors:** Juan Carlos Gomez-Verjan, Marcelino Esparza-Aguilar, Verónica Martín-Martín, Cecilia Salazar-Perez, Cinthya Cadena-Trejo, Luis Miguel Gutierrez-Robledo, José Jaime Martínez-Magaña, Humberto Nicolini, Pedro Arroyo

**Affiliations:** 1Dirección de Investigación, Instituto Nacional de Geriatría, Mexico City 10200, Mexico; ckdnatrejo@hotmail.com (C.C.-T.); parroyo@inger.gob.mx (P.A.); 2Departamento de Investigación en Epidemiología, Instituto Nacional de Pediatría, Mexico City 04530, Mexico; marea.b1@gmail.com (M.E.-A.); iqcebet_9@yahoo.com (C.S.-P.); 3Subdirección de Investigación Médica, Instituto Nacional de Pediatría, Mexico City 04530, Mexico; veromar27@yahoo.com; 4Dirección General, Instituto Nacional de Geriatría, Mexico City 10200, Mexico; lmgutierrez@inger.gob.mx; 5Laboratorio de Genómica de Enfermedades Psiquiátricas y Neurodegenerativas, Instituto Nacional de Medicina Genómica, Mexico City 04809, Mexico; jimy.10.06@gmail.com (J.J.M.-M.); hnicolini@inmegen.gob.mx (H.N.)

**Keywords:** years of schooling, epigenetic age, epigenetics, aging, epigenome-wide association study

## Abstract

Adverse conditions in early life, including environmental, biological and social influences, are risk factors for ill-health during aging and the onset of age-related disorders. In this context, the recent field of social epigenetics offers a valuable method for establishing the relationships among them However, current clinical studies on environmental changes and lifespan disorders are limited. In this sense, the Tlaltizapan (Mexico) cohort, who 52 years ago was exposed to infant malnutrition, low income and poor hygiene conditions, represents a vital source for exploring such factors. Therefore, in the present study, 52 years later, we aimed to explore differences in clinical/biochemical/anthropometric and epigenetic (DNA methylation) variables between individuals from such a cohort, in comparison with an urban-raised sample. Interestingly, only cholesterol levels showed significant differences between the cohorts. On the other hand, individuals from the Tlaltizapan cohort with more years of schooling had a lower epigenetic age in the Horvath (*p*-value = 0.0225) and PhenoAge (*p*-value = 0.0353) clocks, compared to those with lower-level schooling. Our analysis indicates 12 differentially methylated sites associated with the PI3-Akt signaling pathway and galactose metabolism in individuals with different durations of schooling. In conclusion, our results suggest that longer durations of schooling could promote DNA methylation changes that may reduce epigenetic age; nevertheless, further studies are needed.

## 1. Introduction

Biological and social environmental factors influence the development and life expectancy of the average human being [1,2]. The current investigations are centered on the impact of different environmental exposures at different stages of life, including those involved in aging and the manifestation of age-related diseases [3,4,5,6,7]. In this sense, studies have identified a crosslink between early-life adverse conditions and the reduction of life expectancy and “healthspan” [8]. For instance, suffering rheumatic fever and malaria in childhood is associated with a higher prevalence of cardiac disease and increased mortality [9]. Poor socioeconomic status early in life is also predictive of later cognition problems and disability [10]. Another study performed by the Japan Gerontological Evaluation Study Cohort found an association between adverse childhood experiences (parental death, divorce, parental mental illness, family violence, physical abuse, psychological neglect, and psychological abuse) and a greater risk of dementia later in life [11]. In this context, an analysis of the effect of early-life conditions on the disease’s manifestation has two general hypotheses. The antagonistic pleiotropy states that beneficial traits early in life could be detrimental later, due to a decline in the force of natural selection [12]. Furthermore, the Barker hypothesis states that inadequate nutrition in utero may induce an adaptive metabolic program in the fetus that could lead to future disease [13,14]. Some of the most compelling and unfortunate examples of fetal or child detrimental nutrition are historical famines, such as the Chinese Famine or the Hunger Winter [14,15,16,17]. Individuals from those cohorts who were exposed to nutritional deficiency in childhood had a higher risk of age-related illness, such as neurodegenerative diseases, and a meaningful reduction in life expectancy [18,19,20,21].

### 1.1. Social Epigenetics

As mentioned above, as we grow, we are exposed to different adverse environmental stimuli; however, changes in our lifestyle could modify the risk of disease [22,23,24,25]. In this context, the recent field of social epigenetics seems valuable in creating a network of influences to understand the role of the environment and the further development of diseases [26,27,28,29]. Epigenetics mechanisms are genome dynamics that change with time and environmental conditions [30,31]. The most studied epigenetic mechanism in molecular epidemiology concerns analyzing the genome’s DNA methylation patterns (DNAm). This possesses a strong correlation with different diseases, such as metabolic disorders, neurodegenerative diseases, and aging.

Moreover, different research groups worldwide have developed epigenetic clocks, based on data from DNAm patterns [32,33,34,35,36,37], which have proven to be valuable biomarkers for mortality and to be associated with several age-related diseases, such as cancer, diabetes, cardiovascular disease, and frailty, among others [38,39,40]. Although there is little knowledge available on the effect that changes in lifestyle or social environment could have on epigenetic mechanisms, changes in our environment by lifestyle modification, like the number of years of schooling, physical activity, or diet habits, are linked to changes in aging and age-related disorders [32,35].

### 1.2. Tlaltizapan Cohort

In Mexico, between 1966 and 1967, Dr. Joaquin Cravioto recruited all the mothers and children born in a rural town in Tlaltizapan, Morelos, Mexico for study. These individuals were exposed to early adverse conditions, such as low birth weight, low income, poor hygiene conditions at birth and in their homes, and malnutrition. The parents were mainly employed in agriculture; therefore, they were usually exposed to organochlorine-organophosphorus pesticides [41,42]. Such individuals were named the Tlaltizapan cohort and represent an essential source for exploring changes in environmental exposures, not only for the volume of information on them but also because they were born under adverse early-life conditions in a rural community in Mexico. Therefore, in the present study, we aimed to explore the differences in clinical/biochemical and anthropometric variables and DNAm variations between individuals from the Tlaltizapan cohort, 52 years later, and compare them with an urban-raised sample.

## 2. Materials and Methods

### 2.1. Sample Population

Cravioto et al. recruited all children born in around one year (February 1966–February 1967) in Tlaltizapan, with a population of 5637 people. A detailed description of the rationale and methodology of the original study was published in a 1969 Monograph [41,43]. In 2018, we undertook a search of all members of the cohort. Out of 336 births, four were stillborn, 26 died in the first year of life, six died before the age of five, and 16 died before reaching adulthood. The Infant Death Rate was 78.3 per thousand live births. Of the remaining 284 cohort members, 40 were residents of the USA. Ten migrated to the other Mexican States, and we were unable to locate 116. It was possible to track down 116 residents of Tlaltizapan; 36 of them refused to participate in the study, but 82 (50 women and 32 men) gave informed consent to be clinically evaluated at the Instituto Nacional de Geriatría (INGER) in Mexico City. A subgroup comprising 32 birth cohort members was selected for the epigenetic study, based on information regarding family characteristics and living conditions at birth. Additionally, seven healthy middle-class older participants who were born and lived in México City were included as urban-raised individuals. All subjects freely agreed to participate and signed written informed consent forms. The protocol was reviewed and approved by the Bioethics and Research Board of the National Institute of Geriatrics (INGER) and Bioethics and Research Board of the National Institute of Pediatrics (INP) under the number INP-INGER 06/2018. All methods were performed based on the Helsinki declaration and the local Ley General de Salud of Mexico. The information was also protected under the Ley Federal de Proteccion de Datos Personales of Mexico.

### 2.2. Clinical and Anthropometric Measurements

#### 2.2.1. Biochemical Profile

We performed the following standard hematological tests (hemoglobin (g/dL), the percentage of hematocrit, transferring (mg/dL) and the percentage of transferrin saturation, according to Sandler [44], and performed the following standard metabolic tests for glucose (mg/dL) and hemoglobin A1C, according to the World Health Organization (WHO) recommendations [45], and triglycerides (mg/dL), LDL (mg/dL), HDL (mg/dL), cholesterol (mg/dL), creatinine (mg/dL), and uric acid (mg/dL) levels, according to Muñoz et al. [46].

#### 2.2.2. Anthropometric Measurements

We evaluated the following anthropometric measurements: weight (kg), height (cm), knee height (cm), waist circumference (cm), leg circumference (cm), body mass index (BMI—kg/m^2^ height) and percentage of oxygenation, according to the WHO’s recommendations [47].

#### 2.2.3. Body Composition and Physical Performance

For this parameter, we evaluated the appendicular muscular mass (kg/m^2^) and the bone mineral density (T-value) using dual X-ray densitometry (DEXA) [48]. Additionally, we evaluate the short physical performance battery (SPPB): tandem movement (s), unipodal movement (s), walking speed (cm/s), rising from a chair (times/30 s), and maximum grip strength (kg) [49].

#### 2.2.4. Health Status

We evaluated the number of comorbidities, and performed a mini-mental evaluation [50], with depression scale, fear of falls scale [51], frailty scale, falls history, and blood pressure (mmHg).

### 2.3. DNA Methylation

Before clinical and anthropometric evaluation, an EDTA blood sample was collected. According to the manufacturer protocols, DNA was extracted from peripheral blood mononuclear cells using standard methods (Qiagen LLC, Germantown, MD, USA, DNA extraction kit) and was bisulfite-converted using a commercial EZ DNA methylation kit (Zymo Research, Irvine, CA, USA). Then bisulfite converted DNA was processed using the commercial Infinium Methylation EPIC BeadChip (Illumina, San Diego, CA, USA) on the microarray platform located at the Unidad de Microarreglos of the Instituto Nacional de Medicina Genomica (INMEGEN, Mexico City, Mexico), following the manufacturer’s protocol. The EPIC contains 866, 836 CpG sites; this EPIC array has a similar Infinium assay design as the previous 450K array, but it has twice as much coverage of DNA methylation sites (known as CpG sites), which are particularly enriched for promoter and enhancer regulatory regions. Such characteristics provide an increased power and genome coverage to identify novel loci that are relevant for aging and health disparities. The EPIC intensity signals were transformed to *idat* files by the GenomeStudio software (Version 2.0) (Illumina, CA, USA). Then, the *idat* files were processed for quality control under the *ChAMP* package (version 2.22.0). The quality control included the remotion of: (i) 8,250 CpG site with detection *p* > 0.01, (ii) 2,727 < 3 beads in 5% samples, (iii) 2,969 NoCpGs, (iv) 95,778 SNPs [52], (v) 50 multihit [53,54], and (vi) 16,549 CpGs sites found on X and Y chromosomes. We included 12 technical replicates (two per plate) for quality control. We removed those probes with low intensity, as well as those that could introduce biases in performance.

Additionally, for this analysis, we removed the probes for sexual chromosomes so that all subjects could be more easily compared since sexual chromosome DNA methylation at baseline is inherently more variable in females than in males. After quality control, a total of 740,513 CpG sites remained for the subsequent analysis. Then, a matrix of β-values and M-values was generated.

#### 2.3.1. Epigenetic Clocks

Epigenetic clocks were calculated using the *ENmix* package (version 1.28.4) after the ChAMP β-mixture quantile normalization (BMIQ) *.norm* function of the *ChAMP* package [55], using the β matrix. The Hannum, Horvath, and DNA PhenoAge clock values were calculated. The age-acceleration level was also calculated, regressing the epigenetic age calculated by each clock with the chronological age [37]. Pairwise correlations were also calculated between each clock and the chronological age (for example, chronological age was correlated with PhenoAge). The pairwise correlations were performed in both cohorts (Tlatizapan and urban-raised). Pearson correlations were performed, and a *p*-value < 0.05 was considered statistically significant.

### 2.4. Statistical Analysis

#### 2.4.1. Clinical Study, Epigenetic Clocks, and Anthropometric Comparison between the Tlatizapan Cohort and the Urban-Raised Cohort

Continuous variables were represented by means and standard deviations. Meanwhile, categorical variables were represented as both a total and a percentage. The Shapiro–Wilk test evaluated the normal distribution of the variables. All the variables followed the normal distribution. Continuous variables were compared by *t*-test. T-test with Welch correction, and categorical by chi-squared test. A *p*-value < 0.05 was considered statistically significant. These analyses were performed using the R-statistical language (version 3.6.3) [56].

#### 2.4.2. Analysis Only on the Tlatizapan Cohort

##### Epigenetic Clocks

To explore the factors associated with epigenetic aging in the Tlatizapan cohort, we separated the individuals into groups, based on the age acceleration estimated in Section 2.3.1. Individual analyses were performed for each clock (PhenoAge, Horvath, Hannum). Participants were divided based on the mean of their epigenetic age acceleration, i.e., individuals with accelerated aging (those with a higher than zero value for age acceleration) and non-accelerated individuals (those with a value lower than or equal to zero value on age acceleration). Next, we compared all the anthropometric and clinical variables between the groups, as described in Section 2.4.1.

##### Epigenome-Wide Association Analysis

Epigenome-wide analysis (differentially methylated sites) of DNA methylation was evaluated by linear models implemented in the *limma* package (version 3.48.2) [57]. In this analysis, we compared only the Tlatizapan cohort between individuals with high (*n* = 15) and low duration of schooling (*n* = 17). The cut-off of years of schooling was 9 years (Mexico national mean), with high: ≥ 9 years and low: < 9 years. Single-value deconvolution analysis was performed to explore the confounding variables (all the clinical and anthropometric variables explored in the Tlatizapan cohort) with the *champ.SVD* function; no variable was found to be correlated. Batch effects, by slide and array position, were removed by the function *champ.runCombat*. The effect of blood-cell proportion was removed by adjusting the β-values matrix by a reference-based method implemented in the *champ.refbase* function of the *CHAMP* package. For statistical contrast, the β-values were transformed to M-values (better adjustment for linear models), and for reporting the differences in methylation levels, the β-values were used (since they were more biologically interpretable) [58]. A *p*-value < 9 × 10^−8^ was considered statistically significant because this *p*-value threshold adequately controls the false positive rate on the EPIC array [59]. After statistical contrast, we annotated the CpG sites to the nearest gene using the *IlluminaHumanMethylationEPICanno.ilm10b2.hg19* package (version 0.6.0). Gene enrichment analysis was performed using the online tool Webgestalt, using the KEGG pathway database, and an FDR-corrected *p*-value < 0.05 was considered statistically significant [60,61].

### 2.5. Bioethical Considerations

All participating subjects provided written informed consent regarding the study, genetic data, and personal information security. The institutional review boards approved enrollment and consent procedures for this study, given by the Bioethics and Research Board of the National Institute of Geriatrics (INGER) and the Bioethics and Research Board of the National Institute of Pediatrics (INP) under the number INP-INGER 06/2018. All methods employed in the study were performed, following the relevant international guidelines and regulations as well as those from the Ley General de Salud of Mexico. The data are available upon express request addressed to the corresponding author and are currently in safekeeping by INGER, with the approval of the Research and Ethics Committees.

## 3. Results

### 3.1. Differences between the Urban-Raised and the Tlatizapan Cohorts

In the analysis of clinical variables between individuals who were urban-raised and individuals in the Tlaltizapan cohort, we found differences only in the serum total cholesterol level (Table 1).

#### Evaluation of Different Epigenetic Clocks

The lowest correlation with chronological age was found with the Hannum clock. Chronological age was positively correlated with Horvath and negatively correlated with PhenoAge (Figure 1). In the correlation between epigenetic clocks, a higher correlation was between Horvath and Hannum. PhenoAge had a lower correlation with Hannum and Horvath. The epigenetic age of the urban raised was higher than those of the Tlatizapan cohort; nevertheless, this difference could be derived by the higher chronological age of the urban raised individuals.

### 3.2. Analysis of the Tlatizapan Cohort

#### 3.2.1. Effect of Clinical Variables on Individuals with Accelerated Epigenetic Aging

After the evaluation comparison with the urban-raised individuals, we performed a case-only stratified analysis in the Tlatizapan cohort. The individuals were divided into groups based on the mean epigenetic age acceleration for each clock (i.e., a separated analysis for each epigenetic clock), on individuals with accelerated aging (higher than zero on age acceleration), and non-accelerated aging (lower or equal than zero). In this analysis, we only found differences between the number of years of schooling in the groups constructed with Horvath (t = −2.4070, *p*-value = 0.0225) and PhenoAge (t = −2.2110, *p*-value = 0.0353), and no differences were found in Hannum (t = −0.8793, *p*-value = 0.3863). The individuals with no epigenetic age acceleration had longer schooling than those with epigenetic age acceleration (Figure 2). In addition, we found a correlation between the duration of schooling and age acceleration (r = −0.34, *p*-value = 0.0452). We also found that individuals of the Tlatizapan cohort with more years of schooling (>9 years) had a reduction in Horvath clock epigenetic aging (mean = −1.88) compared to the urban-raised individuals (mean = 4.77) (t = −2.7684, *p*-value = 0.0273).

#### 3.2.2. Evaluation of Epigenetic Changes in Individuals with Long and Short Durations of Schooling

Based on the differences found previously, we divided the Tlaltizapan cohort based on the duration of schooling—high (*n* = 15) and low (*n* = 17) number of years in education (high: ≥9 years, low: <9 years). In the analysis of differentially methylated sites between individuals, we did not find any statistically significant signal. Nevertheless, we focused on the 12 top CpG sites (*p*-value < 5 *×* 10^−5^) (Table 2; Appendix A).

The CpG sites that were nominally differentially methylated were found in 6 intergenic regions, five protein-coding regions, and one long non-protein-coding region. Most of the sites (*n* = 8, 66.67%) were found in open-sea regions. Cg03184819 had a higher difference between groups (Log FC = −0.0966), followed by the CpG site found in the transcription starting site of the lactase gene (Log FC = −0.0892); both sites were hypermethylated in the individuals with a higher number of years of schooling. The genes with differentially methylated sites were enriched in the PI3-Akt signaling pathway (hsa04151, adjusted *p*-value = 0.0082, *MAGI1* and *TNC*) and galactose metabolism (hsa00052, adjusted *p*-value = 0.0137, *LCT*).

## 4. Discussion

Social epigenetics is a helpful tool for analyzing the effect of exposures to adverse environments during the first years of life. Hence, in the present report, through such a tool, we analyze a subsample that was part of the Tlaltizapan cohort, which was reported as being exposed to early adverse conditions. Interestingly, according to our results, PhenoAge is not positively correlated with chronological age (in comparison to Horvath’s clock); this could be because this biological clock is more associated with pro-inflammatory and DNA damage routes that are perhaps not yet altered in this population, due to the fact that they are still only in their early fifties.

On the other hand, our findings suggest that individuals with age acceleration in the Tlaltizapan cohort had fewer years of schooling. Such results must be interpreted with caution since correlation is not causation. Nevertheless, this is not the first time that education has been correlated with methylation, and several studies also indicated correlations between methylation patterns and education [32,34,35].

Furthermore, McGuiness et al. [62] suggest that in a cohort of 239 men and women from Scotland, each additional year of education was associated with a 2.4% greater global DNA methylation content. Moreover, van Dongen et al. [32] identified 58 CpG sites associated with educational attainment, and that the methylome level of poorly educated people resembles that of smokers. Additionally, Perng et al. [63] found, in a cross-sectional study of 568 children from Bogota, Colombia that changes in global DNA methylation are associated with different concentrations of micronutrients.

The differences in epigenome-wide analysis (influenced by advanced years of schooling) point to the galactose metabolism’s inactivation. D-galactose’s presence has been a clear indicator of aging by promoting mitochondrial dysfunction and cognitive decline; the mechanism of such changes remains unknown, but epigenetics inactivation could be a fundamental point. On the other hand, the PI3K-Akt pathway was enriched among the associated CpG sites, and this pathway is involved in aging and Alzheimer’s disease [64]. Moreover, this pathway targets the modulation of aging, due to its involvement in skin, muscular, and vascular aging [65].

Among our analysis of epigenome-wide differences between individuals with high or low levels of schooling, the genes with nominal associations were *LCT*, *ASB18*, *MAGI1*, *ANK2*, *TNC*, *LOC100128811.* One of the most interesting findings regarding nutritional status is lactase (*LCT*). A cross-sectional study with 196 malnourished children from Uganda demonstrated that nutritional injury reduces the capacity of the intestinal mucosa to synthesize *LCT* [66]. Moreover, the genetic expression of lactase was reduced in a study with biopsy specimens from 29 malnourished infants [67].

Although our results could generate novel hypotheses for a possible intervention regarding aging, they must be assessed with caution since there are limitations on our study that must be pointed out. For instance, the sample size reduces the statistical power of our analysis, and the age of the individuals is still too young to indicate age-related diseases (52 years old); additionally, more comparisons with a significantly larger number of urban-raised individuals must be carried out. Nevertheless, as one strength, our study carefully characterizes each variable analyzed. Additionally, ancestry is another interesting point to highlight; the analyzed cohort is Mexican-raised. In this sense, Horvath et al. [68] reported that epigenetic age could be different based on ancestry and that estimations on Latin American and Hispanic populations were needed, so our analysis adds information to this particular field.

## 5. Conclusions

In the present study, we subsampled a historical cohort who were recruited more than 52 years ago and who were exposed to several adverse conditions in their early development and in infancy, such as low income, low birth weight, malnutrition, and poor hygienic status. Our results indicate that more years of schooling could reduce epigenetic aging and that PI3K-Akt and galactose metabolism are epigenetically altered. It is noted that further studies with larger sample sizes are needed to validate our results.

## Figures and Tables

**Figure 1 genes-12-01408-f001:**
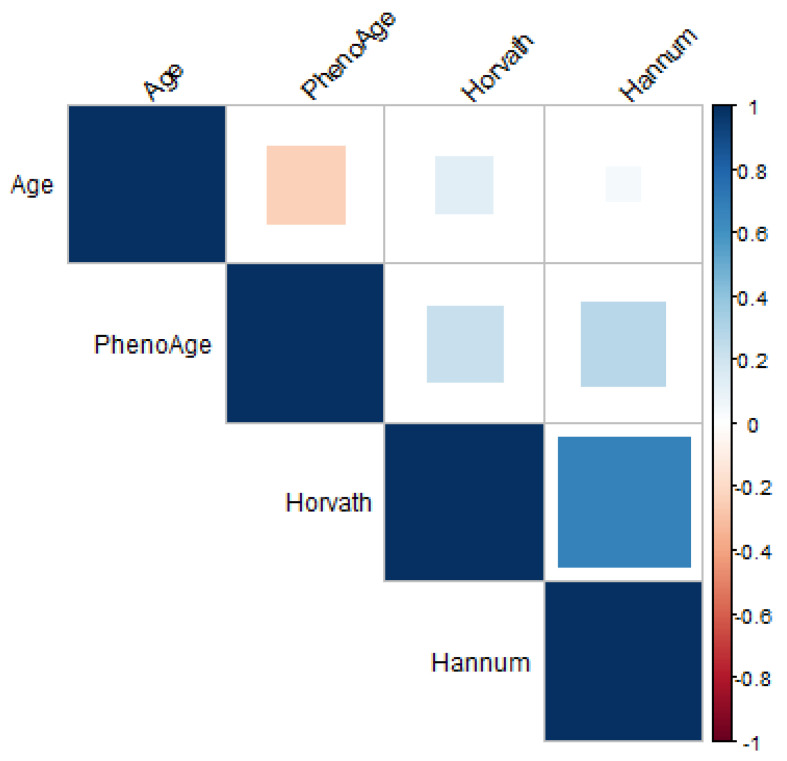
Correlation matrix of the epigenetic clocks and the chronological age. Note: The correlation was performed in both cohorts, Tlaltizapan and urban-raised, and collapsed. Blue colors represent a positive correlation, and red colors represent a negative correlation. The square size signifies the *p*-value of the correlation (the significance is directly proportional to the square size; all the correlations were statistically significant (*p*-value < 0.05).

**Figure 2 genes-12-01408-f002:**
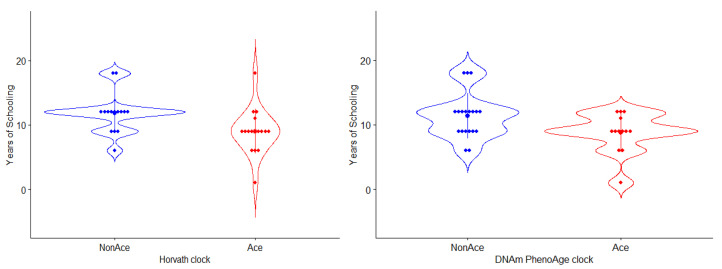
Differences between individuals with epigenetic age acceleration and years of schooling. Note: Blue colors represent individuals with no acceleration on epigenetic clocks, and red represents individuals with accelerated epigenetic aging.

**Table 1 genes-12-01408-t001:** Clinical, biochemical, and epigenetic variables.

	Tlaltizapan Cohort (*n* = 32)	Urban Raised (*n* = 7)	Stat (*p*-Value)
**Age**	**52.26 (0.42)**	**62.41 (0.35)**	**3.98 (0.0105)**
Gender			
Female	19 (59.37)	5 (71.43)	0.03 (0.8690)
Male	13 (40.63)	2 (28.57)	0.03 (0.8690)
Years of schooling	10.31 (3.55)	13.86 (8.99)	1.03 (0.3421)
BMI	29.35 (4.61)	29.12 (4.61)	−0.12 (0.9063)
Visceral fat	3.09 (1.01)	3.17 (0.86)	0.21 (0.8379)
Biochemical variables			
Glucose	108.34 (49.87)	125.43 (69.01)	0.62 (0.5535)
Triglycerides	212.59 (210.05)	230.71 (190.92)	0.22 (0.8281)
**Total Cholesterol**	**196.75 (29.94)**	**236.29 (33.14)**	**2.91 (0.0189)**
HDL	44.07 (12.84)	45.81 (12.09)	0.34 (0.7400)
LDL	111.98 (38.60)	144.30 (44.98)	1.76 (0.1155)
Creatinine	0.64 (0.18)	0.71 (0.14)	1.15 (0.2763)
Uric Acid	5.62 (0.65)	5.21 (1.42)	−1.17 (0.2535)
Prealbumine	24.92 (9.53)	28.82 (5.61)	1.37 (0.1970)
Reactive Protein C	0.40 (0.17)	0.38 (0.08)	−0.55 (0.5935)
Transferrine	247.76 (87.39)	276.40 (42.01)	1.24 (0.2341)
Glycosylated Haemoglobin	6.45 (1.77)	5.73 (1.22)	−1.22 (0.2525)
Fibrinogen	330.75 (50.13)	353.83 (24.79)	1.71 (0.1078)
Iron Fixing	357.06 (60.39)	388.67 (78.15)	0.94 (0.3828)
Mental Health			
Mini-mental	25.56 (2.65)	26.42 (1.81)	1.04 (0.3167)
Depression	6 (18.75)	2 (28.57)	0.00 (0.9472)
Drugs	9 (28.13)	5 (71.43)	5.18 (0.1591)
Tobacco Index	1.50 (5.45)	0.86 (1.18)	−0.61 (0.5477)
Alcohol	28 (87.50)	4 (57.14)	1.83 (0.1763)
Epigenetic Clocks			
**Horvath**	**58.37 (3.61)**	**63.84 (4.56)**	**2.96 (0.0138)**
Horvath acceleration	−0.14 (3.56)	0.71 (4.24)	0.47 (0.6579)
Hannum	56.77 (4.02)	60.16 (5.40)	1.75 (0.1096)
Hannum acceleration	0.05 (3.60)	−0.25 (4.45)	−0.15 (0.8814)
**DNAm PhenoAge**	**49.95 (3.62)**	**54.74 (4.87)**	**2.06 (0.0591)**
PhenoAge acceleration	0.11 (6.64)	−0.58 (4.02)	−0.34 (0.7383)

Note. Bold represents statistically significant differences (not adjusted for the multiple tests).

**Table 2 genes-12-01408-t002:** Epigenome-wide nominal associations between high and low duration of schooling.

^1^ Position	CpG Site	^2^ LogFC	*p*-Value	High Avg	Low Avg	Gene	^3^ Gene Loc	CGI ^4^
2:38496264	cg19269093	−0.0453	2.1563 × 10^−5^	0.8747	0.8294		IGR	OpenSea
2:136595281	cg04750100	−0.0892	2.5258 × 10^−5^	0.4171	0.3279	*LCT*	TSS1500	OpenSea
2:237163447	cg25305153	−0.0779	3.3160 × 10^−5^	0.8126	0.7347	*ASB18*	Body	OpenSea
3:65561644	cg05244979	−0.0510	3.4089 × 10^−5^	0.6869	0.6359	*MAGI1*	Body	OpenSea
4:113970506	cg02815171	0.1165	4.5334 × 10^−5^	0.4218	0.5383	*ANK2*	TSS1500	OpenSea
7:156716133	cg03184819	−0.0907	4.9140 × 10^−5^	0.2529	0.1622		IGR	OpenSea
9:4435234	cg08538646	−0.0717	4.2326 × 10^−5^	0.8585	0.7868		IGR	OpenSea
9:117818174	cg07712264	−0.0555	4.0462 × 10^−5^	0.7059	0.6504	*TNC*	Body	OpenSea
10:25460855	cg15018193	−0.0439	1.1161 × 10^−5^	0.7583	0.7144	*LOC100128811*	Body	Self
11:102124935	cg06458665	−0.0397	2.2954 × 10^−5^	0.8635	0.8237		IGR	OpenSea
12:26451968	cg13537590	−0.0500	1.0817 × 10^−5^	0.7114	0.6614		IGR	OpenSea
22:50710746	cg22416596	−0.0404	4.9783 × 10^−5^	0.6884	0.6481		IGR	Shore

Notes. ^1^ Human genome position of CpG site (GRCh37/hg19). ^2^ LogFC = logarithm of fold change. ^3^ Gene Loc = location of CpG site relative to coding gene. ^4^ CGI = CpG island. ^5^ IGR = Intergenic region.

## Data Availability

The data presented in this study are available on request from the corresponding author. The data are not publicly available due to ethical restrictions.

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
