# Peer review of "Years of Schooling Could Reduce Epigenetic Aging: A Study of a Mexican Cohort"

_genes, 2021, doi:10.3390/genes12091408_

Round 1
Reviewer 1 Report
Years of schooling could reduce epigenetic aging: study of a 2 Mexican cohort
The paper by Gomez-Verjan, examined the association between malnourishment early in life and clinical, anthropometric, biochemical and DNA methylation patterns in a sample of Tlaltizapan cohort. The authors also identified 7 individuals who were considered “Urban raised” and served as controls or comparison cohort. The authors examined both epigenome wide differential DNA methylation at individual CpGs and epigenetic age acceleration measures (based on Horvath, Hannum, and PhenoAge measures).
Introduction
- Please provide more focused and relevant introduction on the role of education in DNA methylation changes.
- The aims of the study mentioned in the abstract and introduction are not consistent. Please revise to clearly state the main analysis of the paper.
Methods
- What was the number of CpGs analyzed after QC?
- Please clarify what confounders were considered in the analysis. It is not clear what covariates were included in the model where years of schooling was found to be significant. Important confounder such as age, sex, and socioeconomic status are important to consider.
Results
- It is not clear what the sample size is for the years of schooling analysis. How many individuals were in each group (high vs. low)?
- The sample size of the cohort is significantly small and raises concerns regarding model fit and reproducibility of the findings.

Author Response
Q1. The paper by Gomez-Verjan examined the association between malnourishment early in life and clinical, anthropometric, biochemical and DNA methylation patterns in a sample of Tlaltizapan cohort. The authors also identified 7 individuals who were considered "Urban raised" and served as controls or comparison cohorts. The authors examined both epigenome wide differential DNA methylation at individual CpGs and epigenetic age acceleration measures (based on Horvath, Hannum, and PhenoAge measures).
Introduction
Q2. Please provide a more focused and relevant introduction on the role of education in DNA methylation changes.
Response = Thank you for your kind suggestion; we modified the introduction to be more precise and more information on the matter. Nevertheless, we decided to add more information on this matter in the discussion rather than the introduction. We believe that such information in this section is more relevant to sustain our analysis than the introduction; please refer to the manuscript.
Q3. The aims of the study mentioned in the abstract and introduction are not consistent. Please revise to clearly state the main analysis of the paper.
Response = Thanks for your valuable comments; we changed the main objectives of the study on abstract and in general in the manuscript to the following:
Therefore, in the present study, we aimed to explore 52 years later for differences in clinical/biochemical/anthropometric variables and epigenetic (DNA methylation) variations between individuals from the Tlatizapan cohort and an urban-raised sample.
Methods
Q4. What was the number of CpGs analyzed after QC?
Response = After Quality Control analysis, a total of 740 513 CpGs remained for subsequent studies.
Q5. Please clarify what confounders were considered in the analysis. It is not clear what covariates were included in the model where years of schooling was found to be significant. Important confounder such as age, sex, and socioeconomic status are important to consider.
Response = We did not use any cofounder on the present analysis since we separated the individuals based on the age acceleration and tested for differences in every clinical and anthropometric variable. We have corrected this point on the manuscript in the methods section as follows:
2.4.2.1. Epigenetic clocks
To explore factors associated with epigenetic aging in the Tlatizapan cohort, we separated the individuals based on the age acceleration estimated in point 2.3.1. Separated analyses were performed for each clock (PhenoAge, Horvath, Hannum). Individuals were divided based on the mean of its epigenetic age acceleration, according to individuals with accelerated aging, those with a higher than zero value on age acceleration, and non-accelerated individuals, those with a lower or equal than zero value on age acceleration. Next, we compared all the anthropometric and clinical variables between the groups, as described on point 2.4.1.
Results
Q6. It is not clear what the sample size is for the years of schooling analysis. How many individuals were in each group (high vs. low)?
Response = Thanks for your commentary; we added the following to the manuscript so that this issue could be more precise:
High (n = 15) and low (n = 17) years scholarship (high: >= 9 years, low: < 9 years).
Q7. The sample size of the cohort is significantly small and raises concerns regarding model fit and reproducibility of the findings.
Response = Thank you for your commentary; we understand your concerns about the sample size, thus in the reviewed version of the manuscript, we include such limitations in the conclusion sections.
Nevertheless, it is essential to highlight that the size of our cohort more than a weakness is a strength in our study since we have valuable clinical data from birth followed 52 years ago in a Mexican rural community, which is quite remarkable because Mexican epigenetic studies are few. The race/ethnicity is quite crucial in the epigenetic age. The longitudinal study of this type of cohort becomes essential to fill the gap in the knowledge about the relation between the early adverse events and the aging process. Finally, this type of genomic clinical study from rare populations is difficult to find because of the migration and deaths.
Nevertheless, several molecular epidemiological studies use small sample sizes already published. Please refer to the following studies as examples:
Merrill SM, Moore SR, Gladish N, Giesbrecht GF, Dewey D, Konwar C, MacIssac JL, Kobor MS, Letourneau NL. Paternal adverse childhood experiences: Associations with infant DNA methylation. Dev Psychobiol. 2021 Aug 1. doi: 10.1002/dev.22174. Epub ahead of print. PMID: 34333774.
Islam SA, Goodman SJ, MacIsaac JL, et al. Integration of DNA methylation patterns and genetic variation in human pediatric tissues help inform EWAS design and interpretation. Epigenetics Chromatin. 2019;12(1):1.
Wang W, Li W, Wu Y, Tian X, Duan H, Li S, Tan Q, Zhang D. Genome-wide DNA methylation and gene expression analyses in monozygotic twins identify potential biomarkers of depression. Transl Psychiatry. 2021 Aug 2;11(1):416. doi: 10.1038/s41398-021-01536-y. PMID: 34341332; PMCID: PMC8329295.
Reviewer 2 Report
The study “Years of schooling could reduce epigenetic aging: study of a Mexican cohort” describes novel DNA methylation results using EPIC arrays on a cohort of subjects with early-life adverse conditions versus 7 urban controls.
There are relevant issues with the manuscript described below:
-
In introduction:
-
When mentioning the Chinese Famine studies, please cite the original reports.
-
The objective, hypotheses of the study are unclear and should be much more clearly stated. I understand that the goal of the study is to first compare Tlaltizapan cohort (n=32) to Urban cohort (n=7) ? (section 3.1) And then, later in the study, to only study the Tlaltizapan cohort? (section 3.2). This should be clearly stated.
-
What are the CLINICAL differences between the Tlaltizapan cohort and the Urban raised cohort? The full clinical information for each sample should be attached as a supplementary table. Also, is there specific information on the particular “early adverse conditions” of the subjects studied from the Tlaltizapan cohort?
-
In Methods, software package versions should be stated
-
In Methods, DNA methylation: which (if any) normalizations were applied to the methylation raw data (IDATs) before doing the differential methylation analyses?
-
In Methods, statistical analyses:
-
If the authors checked normality by Shapiro-Wilk, were all the variables normal? If not, did they use non-parametric tests instead of T-tests?
-
-
a p-value <5e-8 is considered as significant.
-
1) I suppose the authors refer this number to the p-values for the limma differential analysis, not for the other variables in table 1 (because there, they use 0.05). Thus, that line should be moved to be mentioned after the limma analysis methods.
-
2) Why is the arbitrary value of p<5e-8 chosen? the authors should state justification for:
-
1) Using an arbitrary value instead of doing multiple testing correction
-
2) IF they use an arbitrary value, why 5e-8?
-
-
-
In the differential methylation analysis: how did the authors select the covariates for removing their effects? Did they examine PCAs, etc. to see which variables were more important in their data? Why did they not adjust for sex, or age? There is a significant/relevant difference in age between the groups according to Table 1
-
In the differential methylation analysis: why did the authors remove the effects of batch (slide, array) and celltypes from the beta values instead of incorporating their coefficients into the linear models, which is the more common practice?
-
In the differential methylation analysis: did the authors do the testing on beta values or on M-values? It is recommended to perform a logit-transformation of beta-values into M-values to better satisfy the assumptions of the linear models within limma.
-
Are the authors sure that they used the “refBase” package to adjust for celltypes compositon? Or did they use the “champ.refbase” function in the “ChAMP” package?
-
In Methods, DNA methylation epigenetic clocks: did the authors observe correlations of the age acceleration with other variables, for example sex?
-
When the authors describe, in Methods, removing “11 multihit” probes and cite Nordlund, J et al 2013, are they saying that:
-
They applied Norlund J et al methodology: align the probes to the genome, detect multi-aligning probes, remove them
or
-
They used some external list provided by Norlund J et al, to filter multi-hit probes
If it is the 2nd option, could the authors state where in Norlund J et al they provide a list of multihit probes?
Nonetheless, I suggest the authors use a MethylationEPIC-specific probe annotation, such as the cross-hybridising described by Pidsley et al 2016 (Critical evaluation of the Illumina MethylationEPIC BeadChip microarray for whole-genome DNA methylation profiling)
-
In table 1:
-
Are the p-values adjusted for multiple testing for the number of rows of the table?
-
The PhenoAge is labelled as significant but the p-value is > 0.05, not < 0.05. This is a marginal result.
-
-
In Results, epigenetic clocks: the difference between estimated DNAm ages is expected because the Urban group is 10 years older than the Tlaltizapan group. There are NO differences in DNAm age acceleration. Thus the conclusion is that there are no differences in DNAm age acceleration associated to the Tlaltizapan cohort.
-
In Results, figure 1: I suppose the figure describes the raw DNAm values (not age-adjusted). If this is so, the correlation between the chronological age and the DNAm ages is surprisingly low. How do the authors explain this? For example that Hannum or Horvath predictions barely correlate with age?
-
In Results, figure 2: it is a bit difficult to understand the rationale of the comparison. To see if years of schooling is associated to epigenetic age acceleration, the authors should do
-
1) a linear regression test of the form: years schooling ~ age acceleration. And then describe the p-value and sign of the age acceleration coefficient.
-
2) separate the groups into “high” or “low” schooling
-
-
In Results, section 3.2.1.
-
The authors say they use a p-value < 5e-5. But in Methods it is 5e-8. Which one is it?
-
The authors describe gene enrichment analyses. There should be a methodology describing the enrichment analyses (which tests, software, databases were used, etc.)
-
-
In discussion
-
The authors state that “Our findings suggest that such population had an accelerated epigenetic age (Horvath) nevertheless, when analyzed specifically, such acceleration belongs to the less educated individuals.”. But: 1) there is no difference in age acceleration between Tlaltizapan cohort and urban cohort. 2) To be able to say “such acceleration belongs to the less educated individuals”, the authors should compare the acceleration in each education subgroups against the acceleration urban cohort and find significant results.
-

Author Response
Q1. The study “Years of schooling could reduce epigenetic aging: study of a Mexican cohort” describes novel DNA methylation results using EPIC arrays on a cohort of subjects with early-life adverse conditions versus 7 urban controls.
There are relevant issues with the manuscript described below:
In introduction:
Q2. When mentioning the Chinese Famine studies, please cite the original reports.
Response = Thanks for your kind recommendation; we have added the original reports. Please refer to the references section on the main manuscript:
Jowett, A.J. The Growth of China’s Population, 1949-1982 (With Special Reference to the Demographic Disaster of 1960-61). Geogr. J. 1984, 150, 155, doi:10.2307/634995.
Jowett, A.J. The Demographic Responses to Famine: The Case of China 1958 (61). GeoJournal 1991, 23, doi:10.1007/BF00241398.
Q3. The objective, hypotheses of the study are unclear and should be much more clearly stated. I understand that the goal of the study is to first compare Tlaltizapan cohort (n=32) to Urban cohort (n=7) ? (section 3.1) And then, later in the study, to only study the Tlaltizapan cohort? (section 3.2). This should be clearly stated.
Response = Thanks for your valuable comment, we changed the main objectives of the study on abstract and in general in the manuscript to the following, so this could be clearer:
Therefore, in the present study, we aimed to explore 52 years later for differences in clinical/biochemical/anthropometric variables and epigenetic (DNA methylation) variations between individuals from the Tlatizapan cohort and an urban-raised sample.
Q4. What are the CLINICAL differences between the Tlaltizapan cohort and the Urban raised cohort? The full clinical information for each sample should be attached as a supplementary table. Also, is there specific information on the particular “early adverse conditions” of the subjects studied from the Tlaltizapan cohort?
Response = Thanks for your commentary; we include the clinical comparison between the Tlaltizapan cohort and the urban raised in Table 1. Differences are highlighted in bold. Concerning the data availability, this type of clinical assay manages several private information from patients, making it challenging to share the data freely for ethical matters. Nevertheless, data are available upon express request addressed to the corresponding researchers; and under the approval of the Research and Ethics Committees, we added such information to the manuscript. We added a paragraph of information on the introduction about the last question of early adverse conditions, most of which are mainly related to malnutrition and poor hygiene; additionally, we added the original citations to the text, including more information.
Q5. In Methods, software package versions should be stated.
Response = Thanks for your suggestion; we have added the package version to the methods section.
Q6. In Methods, DNA methylation: which (if any) normalizations were applied to the methylation raw data (IDATs) before the differential methylation analyses?
Response = We performed beta-mixture quantile normalization (BMIQ); we have added this to the methods section.
In Methods, statistical analyses:
Q7. If the authors checked normality by Shapiro-Wilk, were all the variables normal? If not, did they use non-parametric tests instead of T-tests?
Response = We use all the variables following a normal distribution; we cleared this matter in the methods section.
Q8. a p-value <5e-8 is considered as significant.
1) I suppose the authors refer to this number to the p-values for the limma differential analysis, not for the other variables in table 1 (because there, they use 0.05). Thus, that line should be moved to be mentioned after the limma analysis methods.
2) Why is the arbitrary value of p<5e-8 chosen? the authors should state justification for:
1) Using an arbitrary value instead of doing multiple testing correction
2) IF they use an arbitrary value, why 5e-8?
Response = Thanks for your observations; we have divided the statistical analysis into clinical and epigenome-wide association studies, which could be more precise. Also, we adjusted the p-value for genome-wide significance for 9e-8, based on the analysis performed by Mansell et al., reporting that this threshold controls for false-positive rate for EPIC arrays.
Mansell, G.; Gorrie-Stone, T.J.; Bao, Y.; Kumari, M.; Schalkwyk, L.S.; Mill, J.; Hannon, E. Guidance for DNA Methylation Studies: Statistical Insights from the Illumina EPIC Array. BMC Genomics 2019, 20, 366, doi:10.1186/s12864-019-5761-7.
Q9. In the differential methylation analysis: how did the authors select the covariates for removing their effects? Did they examine PCAs, etc. to see which variables were more important in their data? Why did they not adjust for sex, or age? There is a significant/relevant difference in age between the groups according to Table 1
Response = Concerning your questions, we perform a single-value deconvolution analysis and correlate the PC with all the clinical variables. We did not find any variable highly correlated with the PC. Therefore, we decided to remove the effect of slide and array, which are a non-biological source of epigenetic variations.
Q10. In the differential methylation analysis: why did the authors remove the effects of batch (slide, array) and celltypes from the beta values instead of incorporating their coefficients into the linear models, which is the more common practice?
Response = We agree with your suggestion. Nevertheless, we decide to follow this approximation, based on the small sample size of our study. Some groups had reported that removal of batch effects on small sample sizes could be a better approximation than those which include covariates in linear models.
Q11. In the differential methylation analysis: did the authors do the testing on beta values or on M-values? It is recommended to perform a logit-transformation of beta-values into M-values to better satisfy the assumptions of the linear models within limma.
Response = Thanks for your suggestion; for statistical contrasts, we used the M-values. For reporting the differences between the methylation levels, we used the beta-values to have more biologically meaningful results and interpretation of the data. We added the following to the manuscript:
For statistical contrast, the beta-values were transformed to M-values (better adjustment for linear models), and for reporting the differences in methylation levels, the beta-values were used (more biologically interpretable) [57].
Q12. Are the authors sure that they used the “refBase” package to adjust for celltypes compositon? Or did they use the “champ.refbase” function in the “ChAMP” package?
Response = Thanks for your observation; we change the manuscript according to: champ.refbase function of the CHAMP package.
Q13. In Methods, DNA methylation epigenetic clocks: did the authors observe correlations of the age acceleration with other variables, for example sex?
Response = Thanks for your comment; the analysis on the Tlatizapan cohort only revealed differences in the years of schooling on individuals with age acceleration.
Q14. When the authors describe, in Methods, removing “11 multihit” probes and cite Nordlund, J et al 2013, are they saying that:
They applied Norlund J et al methodology: align the probes to the genome, detect multi-aligning probes, remove them
or
They used some external list provided by Norlund J et al, to filter multi-hit probes
If it is the 2nd option, could the authors state where in Norlund J et al they provide a list of multihit probes?
Nonetheless, I suggest the authors use a MethylationEPIC-specific probe annotation, such as the cross-hybridizing described by Pidsley et al 2016 (Critical evaluation of the Illumina MethylationEPIC BeadChip microarray for whole-genome DNA methylation profiling)
Response = Thank you for the suggestion; it was pretty valuable. According to your suggestions, we used both annotations (Norlund and Pidsley) and resulted in the removal of 50 multi-hit probes. We added both references for the manuscript.
Q15. In table 1:
Are the p-values adjusted for multiple testing for the number of rows of the table?
Response = Thank you for your observation. The p-values were reported for each variable in our manuscript.
Q16. The PhenoAge is labelled as significant but the p-value is > 0.05, not < 0.05. This is a marginal result.
Response = Thank you for your observation. We corrected this from Table 1. Please refer to the main manuscript.
Q17. In Results, epigenetic clocks: the difference between estimated DAmages is expected because the Urban group is 10 years older than the Tlaltizapan group. There are NO differences in DNAm age acceleration. Thus the conclusion is that there are no differences in DNAm age acceleration associated with the Tlaltizapan cohort.
Response = Thank you for your valuable observation; we agree and changed the results redaction for a more straightforward version: …..The epigenetic age of the urban raised was higher than those of the Tlatizapan cohort; nevertheless, this difference could be derived by the higher chronological age of the urban raised individuals…..
Q18. In Results, figure 1: I suppose the figure describes the raw DNAm values (not age-adjusted). If this is so, the correlation between the chronological age and the DNAm age is surprisingly low. How do the authors explain this? For example, do Hannum or Horvath predictions barely correlate with age?
Response = Thank you for your valuable observation; data represent the correlations between different clocks with age, calculated after normalizing with BMIQ. As you mention, the correlation between ages is pretty interesting; such an effect could be due to the effect of ethnicity/ancestry since it has been reported that the ancestry, mainly Hispanic, could affect the estimations of the epigenetic clocks, mainly Horvath and Hannum (Horvath et al., 2016 doi: 10.1186/s13059-016-1030-0) that is also what makes our study unique since this cohort is Mexican which has few epigenetic databases public at date but also coming from a rural community that originally lived under adverse conditions. We added such information to the discussion section; please refer to the manuscript.
Q19. In Results, figure 2: it is a bit difficult to understand the rationale of the comparison. To see if years of schooling is associated to epigenetic age acceleration, the authors should do
1) a linear regression test of the form: years schooling ~ age acceleration. And then describe the p-value and sign of the age acceleration coefficient.
2) separate the groups into “high” or “low” schooling
Response = Thank you for your commentary; we performed the correlation analysis and reported in results, section 3.2.1. The analysis of high and low years of schooling was our results reported in point 3.2.1.
Also, a correlation between years of schooling and age acceleration was found (r = -0.34, p-value = 0.0452).
Q20. In Results, section 3.2.1.
The authors say they use a p-value < 5e-5. But in Methods, it is 5e-8. Which one is it?
Response = Thanks for your observation; we used a p-value of 5e-8, we have corrected such an issue in the manuscript.
Q21. The authors describe gene enrichment analyses. There should be a methodology describing the enrichment analyses (which tests, software, databases were used, etc.)
Response = Thank you; we have described such methodology in the manuscript section entitled gene enrichment analysis as follows:
We annotated the CpG sites to the nearest gene using IlluminaHumanMethylationEPICanno.ilm10b2.hg19 package (version 0.6.0). Gene enrichment analysis was performed using the online tool Webgestalt, using the KEGG pathway database, and an FDR corrected p-value < 0.05 was considered statistically significant.
In discussion
Q22. The authors state that “Our findings suggest that such population had an accelerated epigenetic age (Horvath) nevertheless when analyzed specifically, such acceleration belongs to the less educated individuals.”. But: 1) there is no difference in age acceleration between Tlaltizapan cohort and urban cohort. 2) To be able to say “such acceleration belongs to the less educated individuals”, the authors should compare the acceleration in each education subgroups against the acceleration of the urban cohort and find significant results.
Response = Thanks for your commentary; we change the redaction of that section to:
...Our findings suggest that individuals with age acceleration in the Tlatizapan cohort had fewer years of schooling…., so this could be clearer; additionally, we performed the suggested comparisons and found statistically significant results between the individuals of the Tlatizapan cohort with high years of schooling and the urban-raised and added the following to the results section:
We also found that individuals of the Tlatizapan cohort with high years of schooling (> 9 years) had a reduction in Horvath epigenetic aging (mean = -1.88) compared to the urban-raised individuals (mean = 4.77) (t = -2.7684, p-value = 0.0273), differences not found on individuals with low years of schooling (t = -1.3568, p-value = 0.2243).
Reviewer 3 Report
The ms. “Years of schooling could reduce epigenetic aging: study of a 2 Mexican cohort” by Gomez-Verjan et al. reports the analysis DNA methylation in a small sample of subjects very well defined for their early years of life. The analysis is quite interesting and suggestive. The main problems of the ms. are the English form and the small sample.
The first problem can be solved by a revision of a native English speaker, to avoid the many mistakes disseminated in the ms,
As to the problem of the sample size, which is acknowledged the Authors, it is clear that this undermine the significance of the study and of the results. Then the conclusions should be downtoned.
Another point that should be highlighted is that is not clear the analysis of specific genes. It should be better explained.
Finaly I noticed, in the autothorship list, that Authors with the same affilition do not share the same number.
Author Response
Q1. The ms. “Years of schooling could reduce epigenetic aging: study of a 2 Mexican cohort” by Gomez-Verjan et al. reports the analysis DNA methylation in a small sample of subjects very well defined for their early years of life. The analysis is quite interesting and suggestive. The main problems of the ms. are the English form and the small sample.
Q2. The first problem can be solved by a revision of a native English speaker, to avoid the many mistakes disseminated in the ms.
Response = Thanks for your kind suggestion; we sent our manuscript to a native speaker and corrected all the miscues; please refer to the manuscript.
Q3. As to the problem of the sample size, which is acknowledged by the Authors, it is clear that this undermines the significance of the study and of the results. Then the conclusions should be downtoned.
Response = Thanks for your valuable observation; we modified the conclusion and discussion so the tone in the manuscript could be more cautious.
Q4. Another point that should be highlighted is that it is not clear the analysis of specific genes. It should be better explained.
Response = Thanks for your suggestion, we changed redaction of the starting paragraph in the discussion, so this could be explained:
.. Among our analysis of epigenome-wide differences between individuals with high or low years of schooling, the genes with nominal associations were LCT, ASB18, MAGI1, ANK2, TNC, LOC100128811. One of the most interesting with nutritional status is Lactase (LCT)....
Q5. Finally, I noticed, in the authorship list, that Authors with the same affiliation do not share the same number.
Response = Thanks for this observation; we have corrected this issue.
Round 2
Reviewer 2 Report
The authors have made extensive changes to the work. The reviewer appreciates the efforts made in improving the manuscript. The methods section is much clearer now. Nonetheless, there are still key aspect of the work which are confusing:
- In the abstract, the following sentence is confusing:
Our analysis indicates 12 sites associated with the PI3-Akt signalling pathway and galactose metabolism were differentially methylated regions.
It says “12 sites” but later, “methylated regions”. They are either sites (single CpGs) or regions, not both.
Also, it does not say between which groups these were differentially methylated. Reading the paper, these appear to be differentially methylated sites between high and low schooling within the Tlatizapan cohort.
- In introduction, this two sentences appear to be duplicated or redundant and should be collapsed:
Moreover, different research groups worldwide have developed epigenetic clocks with data from DNAm studies [32–34]. Additionally, different research groups worldwide have developed epigenetic clocks based on DNAm patterns [36,37]
- In the author response to Q3 of the first review, the authors state that they have “changed the main objectives of the study on the abstract and in general in the manuscript…”, so now their objective is:
Therefore, in the present study, we aimed to explore 52 years later for differences in clinical/biochemical/anthropometric variables and epigenetic (DNA methylation) variations between individuals from the Tlatizapan cohort and an urban-raised sample.
However, their abstract does NOT mention any urban-raised cohort, and the introduction explicitly avoids mentioning this cohort, in the following phrase:
Therefore, in the present
study, we aimed to explore 52 years later for differences in
clinical/biochemical/anthropometric variables and DNAm variations between individuals
from the Tlatizapan cohort.
In Results 3.1:
- In Table 1, there should be a p-value indicating if there are (or not) differences in sex proportions between Tlaltizapan and Urban Raised. It is the only clinical variable with a missing p-value.
- In Table 1 Note, please explicitly state that these p-values are not adjusted for multiple testing.
- In the first version, the differences in DNAm age accelerations between Tlaltizapan and Urban were described, but now there is no information on the text. These were results which should be commented. Right now the section is very short. (There appear to be no differences between the 2 cohorts, and this is OK and should be described). Also comment here at least a bit on the correlation results from Figure 1.
- Figure 1 on the whole is not too informative. It shows correlations between DNAm ages and chronological age. Is it a correlation for all samples (Tlaltizapan and Urban, collapsed) or only one cohort?
- In the author response to Q17 of the first review, the authors state that they have changed the results redaction to “The epigenetic age of the urban raised was higher than those of the Tlatizapan cohort; nevertheless, this difference could be derived by the higher chronological age of the urban raised individuals”. However I do not seem to find this phrase anywhere.
In Results 3.2.2:
- The authors write “CpG’s”. It is “CpGs”.
- In 3.2.2, since the 12 CpG sites found are non-significant (because your threshold is 9e-08), I would not write “12 CpG sites with nominal significance”, but rather: “we focused on the 12 top CpG sites (p-value < 5e-05)”
- In 3.2.2, the authors say that “CpG site found in the transcription starting site of the Lactase gene (methylation delta = -0.0892)”. However, this number corresponds to “logFC” in the Table 2. Is is logFC, or is it “delta”? Same thing for Cg03184819.
Author Response
The authors have made extensive changes to the work. The reviewer appreciates the efforts made in improving the manuscript. The methods section is much clearer now. Nonetheless, there are still key aspect of the work which are confusing:
Q1. In the abstract, the following sentence is confusing:
Our analysis indicates 12 sites associated with the PI3-Akt signalling pathway and galactose metabolism were differentially methylated regions.
It says “12 sites” but later, “methylated regions”. They are either sites (single CpGs) or regions, not both.
Also, it does not say between which groups these were differentially methylated. Reading the paper, these appear to be differentially methylated sites between high and low schooling within the Tlatizapan cohort.
Response : Thanks for your observation, we changed the redaction in the abstract, so this could be clearer.
Q2. In introduction, this two sentences appear to be duplicated or redundant and should be collapsed:
Moreover, different research groups worldwide have developed epigenetic clocks with data from DNAm studies [32–34]. Additionally, different research groups worldwide have developed epigenetic clocks based on DNAm patterns [36,37]
Response: Thanks for your suggestion, we have removed this duplicate sentence.
Q3. In the author response to Q3 of the first review, the authors state that they have “changed the main objectives of the study on the abstract and in general in the manuscript…”, so now their objective is:
Therefore, in the present study, we aimed to explore 52 years later for differences in clinical/biochemical/anthropometric variables and epigenetic (DNA methylation) variations between individuals from the Tlatizapan cohort and an urban-raised sample.
However, their abstract does NOT mention any urban-raised cohort, and the introduction explicitly avoids mentioning this cohort, in the following phrase:
Therefore, in the present study, we aimed to explore 52 years later for differences in clinical/biochemical/anthropometric variables and DNAm variations between individuals from the Tlatizapan cohort.
Response : Thanks for your suggestion, we modify the text so this issue could be cleared.
Q4. In Results 3.1:
In Table 1, there should be a p-value indicating if there are (or not) differences in sex proportions between Tlaltizapan and Urban Raised. It is the only clinical variable with a missing p-value.
In Table 1 Note, please explicitly state that these p-values are not adjusted for multiple testing.
Response: Thanks for your suggestions, we have added the p-value indicated in Table 1.
Q5. In the first version, the differences in DNAm age accelerations between Tlaltizapan and Urban were described, but now there is no information on the text. These were results which should be commented. Right now the section is very short. (There appear to be no differences between the 2 cohorts, and this is OK and should be described). Also comment here at least a bit on the correlation results from Figure 1.
Figure 1 on the whole is not too informative. It shows correlations between DNAm ages and chronological age. Is it a correlation for all samples (Tlaltizapan and Urban, collapsed) or only one cohort?
In the author response to Q17 of the first review, the authors state that they have changed the results redaction to “The epigenetic age of the urban raised was higher than those of the Tlatizapan cohort; nevertheless, this difference could be derived by the higher chronological age of the urban raised individuals”. However I do not seem to find this phrase anywhere.
Response : Thanks for your suggestions, we have added information concerning figure 1 and missing text.
Q6. In Results 3.2.2:
The authors write “CpG’s”. It is “CpGs”.
In 3.2.2, since the 12 CpG sites found are non-significant (because your threshold is 9e-08), I would not write “12 CpG sites with nominal significance”, but rather: “we focused on the 12 top CpG sites (p-value < 5e-05)”
In 3.2.2, the authors say that “CpG site found in the transcription starting site of the Lactase gene (methylation delta = -0.0892)”. However, this number corresponds to “logFC” in Table 2. Is is logFC, or is it “delta”? Same thing for Cg03184819.
Response: Thanks for your observations, we have corrected all these miscues on the manuscript.